# Valpalf^®^: A New Nutraceutical Formulation Containing Bovine Lactoferrin That Exhibits Potentiated Biological Activity

**DOI:** 10.3390/ijms25168559

**Published:** 2024-08-06

**Authors:** Luigi Rosa, Giusi Ianiro, Antonella Niro, Giovanni Musci, Rosalba Paesano, Antimo Cutone, Piera Valenti

**Affiliations:** 1Department of Public Health and Infectious Diseases, Sapienza University of Rome, 00185 Rome, Italy; luigi.rosa@uniroma1.it (L.R.); piera.valenti@uniroma1.it (P.V.); 2Department of Biosciences and Territory, University of Molise, 86090 Pesche, Italy; giusy.ianiro@unimol.it (G.I.); a.niro2@studenti.unimol.it (A.N.); musci@unimol.it (G.M.); 3Microbo s.r.l., 00153 Rome, Italy; info@microbosrl.it

**Keywords:** bovine lactoferrin, Valpalf^®^, iron chelation, proteolysis resistance, antioxidant activity, anti-inflammatory activity, anemia of inflammation

## Abstract

As a nutraceutical, bovine lactoferrin (bLf), an iron-binding glycoprotein involved in innate immunity, is gaining elevated attention for its ability to exert pleiotropic functions and to be exceptionally tolerated even at high dosages. Some of bLf’s activities, including its anti-inflammatory and antioxidant, are tightly linked to its ability to both chelate iron and enter inside the cell nucleus. Here, we present data about Valpalf^®^, a new formulation containing bLf, sodium citrate, and sodium bicarbonate at a molar ratio of 10^−3^. In the present study, Valpalf^®^ exhibits superior iron-binding capacity, resistance to tryptic digestion, and a greater capacity to accumulate into the nucleus over time when compared to the native bLf alone. In agreement, Valpalf^®^ effectively reduces interleukin(IL)-6 levels in lipopolysaccharide-stimulated macrophages and modulates the expression of antioxidant enzymes, such as superoxide dismutase 1 and 2, in phorbol-12-myristate-13-acetate-stimulated monocytes. Of note, this potentiated bioactivity was corroborated in a retrospective study on the treatment of anemia of inflammation in hereditary thrombophilic pregnant and non-pregnant women, demonstrating that Valpalf^®^ improves hematological parameters and reduces serum IL-6 levels to a higher extent than bLf alone.

## 1. Introduction

In the vast landscape of multifunctional proteins crucial to human health, lactoferrin (Lf) emerges as one of the most fascinating and versatile protagonists. This glycoprotein, belonging to the transferrin family and found in various biological secretions such as breast milk and bodily fluids, not only provides protection to infants but also carries out a myriad of biological functions that extend far beyond its initial role as an iron-chelating protein possessing antibacterial activity [1,2].

The molecular structure of Lf is a fascinating example of biochemical evolution, featuring two symmetric lobes enclosing the iron-binding sites connected by a short α-helical linker peptide. This bilobal structure facilitates Lf’s crucial role in iron binding and its cationic feature constitutes the basis for a wide range of protein-protein and protein-molecule interactions essential for its multifaceted functions. Within each lobe, a single iron-binding site is situated within a deep cleft formed by two subdomains, the N-lobe and the C-lobe [3]. The iron-binding sites in Lf show a high affinity for ferric ions (Fe^3+^), which is crucial for its biological functions. This affinity of Lf for iron is very high (about 260 times higher than that of serum transferrin) with a constant of about 10^−20^ M [4]. In physiological conditions, Lf is partially saturated with iron (15–20%) (native-Lf) and exhibits a salmon-pink color. Iron-depleted Lf, with less than 5% saturation, is referred to as apo-Lf, while iron-saturated Lf, with over 90% saturation, is called holo-Lf [5].

Iron binding by Lf follows a sequential mechanism. Initially, ferric ions specifically bind with high affinity to four amino acid ligands—including aspartate, histidine, and two tyrosines—which coordinate, primarily in the N-lobe, the ferric ions through bidentate interactions [4,6]. Once bound, ferric ions remain inside the closed form of Lf. Conversely, iron release from Lf is triggered by protonation events at low pH, leading to ligand dissociation and conformational changes that result in the separation of domains and thus forming the open form of Lf. Therefore, Lf exhibits dynamic structural flexibility, allowing it to undergo reversible structural modifications (from open to closed form) in response to iron chelation.

Studies have shown that the iron saturation rate positively impacts Lf’s resistance to proteolysis [7,8], thus allowing a large percentage of the dietary glycoprotein to remain undigested when it reaches the duodenum, the main entry site for Lf into the lymphatic and circulatory system [9,10]. Lf can be internalized via receptor-mediated endocytosis, facilitated by interactions with specific cell surface receptors including the human Lf receptor (rhLf) [11,12]. Once internalized, Lf can resist proteolytic degradation within the cellular environment, thereby preserving its characteristics. Lf exploits this wide range of receptors to activate different cellular responses, either by triggering intracellular signaling or by direct cell entrance through clathrin-mediated endocytosis and subsequent translocation into the nucleus [13]. Once in the nucleus, Lf binds to genes coding for proinflammatory cytokines, thus exerting its anti-inflammatory functions [14,15]. While both native (nat)- and holo-Lf may serve dissimilar functions [5,16], variations in their stability could render one form more efficacious than the other depending on the context [17]. The fully iron-saturated form exhibits greater stability than the partially saturated native counterpart, thereby potentially enhancing its bioactive functions over time. However, low-saturated Lf can still take part in the chelation of unbound iron, which can trigger/exacerbate inflammatory and pro-oxidant processes [2].

Within this frame, Lf’s ability to retain iron at low pH [2], its extracellular and intracellular stability, and its interaction with proinflammatory cytokine genes contribute to its multifaceted protective role in physiological processes, including defense against microbial infections, inflammation, and antioxidant activity [18,19].

Various studies demonstrated that Lf exerts significant anti-microbial activity through the sequestration of essential ferric ions pivotal for bacterial pathogen growth or intracellular viral replication. Moreover, Lf carries out anti-adhesive and anti-invasive activities through its direct interaction with both bacterial/fungal membranes and viral particles as well as the modulation of immune responses and the decrease in proinflammatory cytokines synthesis to combat infections and inflammations [20,21,22].

Numerous in vitro [15,23,24,25], in vivo [26,27,28], and clinical [29,30] studies have highlighted the potent anti-inflammatory properties of bovine Lf (bLf) through various mechanisms, including the suppression of pro-inflammatory cytokine gene activation [29,30] and the modulation of immune responses [18].

In addition, bLf antioxidant activity is intrinsically linked to its ability to sequester ferric ions, thereby preventing the formation of reactive oxygen species (ROS) and reactive nitrogen species (RNS) [2,31]. By binding ferric ions, bLf limits their participation in Fenton and Haber-Weiss reactions, which generate harmful hydroxyl radicals. Additionally, Lf upregulates the expression and activity of antioxidant enzymes such as glutathione peroxidase (GPx) and superoxide dismutase (SOD), bolstering the cellular antioxidant defense system [32]. A recent investigation by our group sheds light on the impact of iron saturation on Lf’s ability to counteract oxidative stress and neurotoxicity triggered by HIV-1 Tat protein [33], emphasizing the importance of evaluating iron saturation status to fully comprehend Lf’s therapeutic efficacy.

The current increasing recognition of Lf as a valuable nutraceutical is expanding its scope beyond traditional uses. Its incorporation into dietary supplements, functional foods, and pharmaceutical formulations offers promising avenues for improving health and facing various diseases. In this regard, new formulations should be useful to potentiate Lf multi-functionalities, including its ability to chelate iron.

In the present study, a new patented formulation of bLf, a mixture of the bLf, sodium citrate, and sodium bicarbonate registered as Valpalf^®^ has been characterized at both the physico-chemical and functional levels.

Here, we reported the in vitro studies showing that Valpalf^®^ has a higher efficacy than bLf alone in iron chelation, resistance to proteolysis, cellular internalization, and anti-inflammatory and antioxidant activities. Surprisingly, Valpalf^®^ is particularly more effective than bLf alone in treating anemia of inflammation (AI) in hereditary thrombophilic pregnant and non-pregnant women.

## 2. Results

### 2.1. Iron Chelation Ability of Valpalf^®^

As a proteic nutraceutical, the efficacy of bLf relies on its physico-chemical characteristics, including its ability to chelate iron [2]. In order to investigate the possible difference in iron-binding activity between Valpalf^®^ and nat-bLf, an iron titration assay was carried out. Obviously, holo-bLf, being completely iron-saturated, was incapable of chelating further ferric ions. Therefore, 10 mg of nat-bLf (corresponding to a final concentration of 1.25 × 10^−4^ M) were dissolved in 1 mL of different solutions potentially able to influence its iron-binding ability, namely sodium chloride (1.5 × 10^−1^ M), sodium citrate (4.0 × 10^−2^ M), and sodium bicarbonate (8.5 × 10^−2^ M), whereas Valpalf^®^ was solubilized in distilled water. The results of the assay are presented in Table 1.

In the initial measurements at 468 nm, the optical density (OD) values for bLf in different solutions were as follows: 0.050 in saline solution (9.26% iron saturation), 0.052 in sodium citrate solution (9.63% iron saturation), 0.051 in sodium bicarbonate solution (9.44% iron saturation), and 0.050 for Valpalf^®^ solubilized in distilled water (9.26% iron saturation).

Out of 10 mg of bLf for each solution, subsequent titration with repeated additions of ferric ions (FeCl_3_ 0.01 M) revealed the following protein amounts active in chelating iron:-bLf in saline solution: 2.8 mg (maximum absorbance at 468 nm OD 0.151)-bLf in sodium citrate solution: 2.9 mg (maximum absorbance at 468 nm OD 0.157)-bLf in sodium bicarbonate solution: 2.9 mg (maximum absorbance at 468 nm OD 0.158)-Valpalf^®^: 10 mg (maximum absorbance at 468 nm OD 0.540)

The iron titration assay showed that the iron saturation degree is approximately 28–29% for all solutions of nat-bLf, with the notable exception of Valpalf^®^, which is the only one to reach a 100% degree of iron saturation. This demonstrates its higher iron binding capacity compared to nat-bLf dissolved in sodium citrate or sodium bicarbonate alone, at the same concentrations utilized in Valpalf^®^ where sodium citrate and bicarbonate are added simultaneously.

In light of these results, all further experiments have been conducted by comparing nat-bLf, holo-bLf, and Valpalf^®^ in various biological activities.

### 2.2. Tryptic Resistance of Valpalf^®^

Another pivotal feature influencing the bioactivity of bLf is the resistance to proteolysis. For this purpose, the tryptic digestion of nat-bLf, holo-bLf, and Valpalf^®^ was carried out. The different Lfs were incubated with trypsin at 37 °C, and aliquots were taken at different times of incubation (0, 10′, 30′, 1 h, 3 h, and 6 h). The protein digests were analyzed by SDS-PAGE and Coomassie blue staining (Figure 1).

All three bLfs initially showed a single intact band at about 80 kDa (Figure 1A). Holo-bLf seemed to possess a slightly higher electrophoretic mobility with respect to nat-bLf and Valpalf^®^. This result may be due to the conformational changes of holo-bLf (completely iron-saturated) with respect to nat-bLf and Valpalf^®^ (partially iron-saturated) [4]. After trypsin digestion, all bLfs showed a peculiar pattern of distinct bands whose intensities were directly related to the time of incubation. Nat-bLf generally displays resistance to proteolytic degradation, which was confirmed here by the only partial digestion of the intact polypeptide even after 6 h of digestion. As expected, Holo-bLf showed a more pronounced resistance to proteolysis than the native form. In general, Valpalf^®^ was more resistant than the nat-bLf, showing a pattern of degradation more similar to the iron-saturated glycoprotein. The statistical analysis highlighted a significant difference between Valpalf^®^ and nat-bLf after 10, 30, and 60 min of digestion (Figure 1).

### 2.3. Lactoferrins Internalization and Subcellular Localization

BLf has been shown to exert multiple functions by triggering either intracellular signaling cascades or receptor-mediated endocytosis and translocation to the cell nucleus. However, the formulation of bLf can influence its stabilization, solubility, iron chelation, and, consequently, its internalization and activity. Therefore, the subcellular localization of nat-Lf, holo-Lf, and Valpalf^®^ was assessed in intestinal Caco-2 cells (after 6 and 24 h of incubation) to mimic the oral administration of Valpalf^®^ (Figure 2).

Following 6 h of incubation, nat-bLf, holo-bLf, and Valpalf^®^ were effectively internalized in Caco-2 cells (Figure 2A), with higher levels in the cytoplasmic compartments with respect to the nucleus. In particular, holo-bLf seemed to accumulate more rapidly than both nat-bLf and Valpalf^®^. After 24 h of treatment, all bLfs accumulate into the nuclear fraction, with Valpalf^®^ showing higher levels than the holo counterpart in both cytosolic and nuclear compartments (Figure 2B).

Overall, these data suggest that Valpalf^®^ can enter the cell in a manner similar to the native protein, but showing greater stability and higher capacity to accumulate into the nucleus over time when compared to both nat-bLf and holo-bLf.

### 2.4. Anti-Inflammatory Activity of Valpalf^®^

To investigate the effect of Valpalf^®^ on the inflammatory response, the expression level of IL-6 was measured in the supernatants of differentiated THP-1 cells challenged with lipopolysaccharide (LPS). THP-1 cells were stimulated with 1 µg/mL LPS in the presence or absence of 100 µg/mL nat-bLf, holo-bLf, or Valpalf^®^. As shown in Figure 3, LPS induced a significant increase in IL-6 levels, which was counteracted by treatments with all bLfs, although at different extents. Interestingly, Valpalf^®^ was more effective than nat-bLf and holo-bLf in reducing IL-6 levels.

### 2.5. Antioxidant Activity of Valpalf^®^

As inflammation is closely associated with intracellular oxidative stress, we proceeded to analyze bLf antioxidant activity. Upon stimulation with phorbol 12-myristate 13-acetate (PMA), THP-1 cells differentiate into mature macrophages, expressing pro-inflammatory markers and regulating anti-oxidant enzymes [34]. Therefore, the protein expression of the superoxide dismutases SOD-1 and SOD-2 was assessed through Western blot analysis (Figure 4).

The expression of SOD-1 was significantly higher in undifferentiated cells treated with only nat-bLf when compared to the control, while holo-bLf and Valpalf^®^ were ineffective (Figure 4A). On the other hand, a slight reduction in SOD-1 expression was observed upon PMA treatment, which was significantly counteracted by pre-treatments with both holo-bLf and Valpalf^®^.

In PMA-untreated cells, the expression of SOD-2 did not seem to be influenced by the treatments with different bLfs (Figure 4B); however, differentiated cells showed a significant increase in SOD-2 when compared to the control. This effect was significantly counteracted by nat-bLf and, to a higher extent, by Valpalf^®^, but not by holo-bLf. This suggests an adaptive antioxidant response during the differentiation process and the differential impact of bLfs on SOD-2 expression underscores their distinct modulatory effects on antioxidant pathways, with nat-bLf and Valpalf^®^ able to potentiate the antioxidant activity through direct chelation of free iron.

Overall, these findings highlight the nuanced effects of different bLfs formulations on antioxidant enzyme expression, providing valuable insights into their potential roles in cellular redox regulation. In particular, the effects of Valpalf^®^ treatment on antioxidant enzymes resemble those exerted by nat-bLf, which is already being described as the more efficient form in preventing and counteracting redox imbalance [33].

### 2.6. Retrospective Study: Efficacy of Valpalf^®^ in the Treatment of Anemia of Inflammation in Hereditary Thrombophilic Pregnant and Non-Pregnant Women

The efficacy of bLf in treating AI in HT women, either pregnant or non-pregnant, has been widely demonstrated in different studies [29,30]. To validate and corroborate the results in vitro, we compared the efficacy of the administration of 200 mg twice a day of bLf alone or Valpalf^®^ in improving the hematological parameters of HT pregnant and non-pregnant women suffering from AI (Figure 5).

A total of 67 pregnant and non-pregnant women suffering from HT were involved in the study: 35/67 HT pregnant women and 37/67 HT non-pregnant women were treated before meals with 200 mg twice a day for 30 days with bLf, while 32/67 HT pregnant women and 30/67 HT non-pregnant women were treated before meals with 200 mg twice a day for 30 days with Valpalf^®^ (Figure 5).

The demographic data and the baseline levels of hematological parameters are reported in Table 2 and Table 3 for HT pregnant and non-pregnant women, respectively.

The results are reported in Figure 6 for HT pregnant women and in Figure 7 for HT non-pregnant women.

In HT pregnant women affected by AI, after 30 days of treatment, both bLf and Valpalf^®^ induced a significant increase in tested parameters (Figure 6). In particular, bLf treatment induced an increase in the levels of red blood cells (RBCs) (4248 ± 147.6 × 10^3^ cells/mL, +16.7 ± 4.2%, *p* < 0.0001), hemoglobin (Hb) (12.8 ± 0.5 g/dL, +24.6 ± 8.1%, *p* < 0.0001), total serum iron (TSI) (89 ± 7.8 μg/dL, +150.1 ± 40.4%, *p* < 0.0001), and serum ferritin (sFtn) (25.3 ± 3.7 ng/mL, +173.7 ± 119.7%, *p* < 0.0001), along with a parallel decrease in IL-6 levels (43.3 ± 6.6 pg/mL, −51.2 ± 8.7%, *p* < 0.0001), compared to their respective baseline values (Table 2).

Valpalf^®^ increased the levels of RBCs (4539 ± 153.3 × 10^3^ cells/mL, +25.2 ± 5.9%, *p* < 0.0001), Hb (13.5 ± 0.4 g/dL, +28.7 ± 7.4%, *p* < 0.0001), TSI (120.1 ± 8.2 μg/dL, +237.6 ± 62.2%, *p* < 0.0001), and sFtn (27.2 ± 4.9 ng/mL, +190.6 ± 112.2%, *p* < 0.0001), along with a parallel decrease in IL-6 levels (35.6 ± 4.1 pg/mL, −61.5 ± 4.8%, *p* < 0.0001), compared to their respective baseline values (Table 2).

Of particular note, Valpalf^®^ demonstrated a significantly greater increase in RBCs, Hb, and TSI (all *p* < 0.0001), except for sFtn, accompanied by a more pronounced decrease in IL-6 (*p* < 0.001) compared to bLf treatment alone (Figure 6).

The same trend was observed in HT non-pregnant women suffering from AI, where treatments with both bLf alone and Valpalf^®^ increased the levels of tested parameters (Figure 7). BLf treatment induced an increase in the levels of RBCs (4292 ± 131.1 × 10^3^ cells/mL, +17.7 ± 4.4%, *p* < 0.0001), Hb (12.8 ± 0.5 g/dL, +22.9 ± 9.2%, *p* < 0.0001), TSI (97.1 ± 6.3 μg/dL, +161.7 ± 37.6%, *p* < 0.0001), and sFtn (30.8 ± 3.3 ng/mL, +153.0 ± 49.1%, *p* < 0.0001), along with a parallel decrease in IL-6 levels (21.5 ± 3.5 pg/mL, −60.1 ± 6.7%, *p* < 0.0001), compared to their respective baseline values (Table 3).

Valpalf^®^ increased the levels of RBCs (4521 ± 145.1 × 10^3^ cells/mL, +24.6 ± 5.8%, *p* < 0.0001), Hb (14.0 ± 0.5 g/dL, +30.9 ± 7.1%, *p* < 0.0001), TSI (103.1 ± 6.5 μg/dL, +172.9 ± 27.3%, *p* < 0.0001), and sFtn (32.8 ± 3.7 ng/mL, +204.2 ± 65.5%, *p* < 0.0001), along with a parallel decrease in IL-6 levels (14.9 ± 4.5 pg/mL, −73.9 ± 4.7%, *p* < 0.0001), compared to their respective baseline values (Table 3).

Even in this group, Valpalf^®^ increased RBCs, Hb (both with *p* < 0.0001), TSI (*p* < 0.001), and IL-6 (*p* < 0.01), but not sFtn, at a greater magnitude with respect to bLf treatment (Figure 7). Importantly, no women administered with either bLf alone or Valpalf^®^ experienced any side effects during the period of treatment.

## 3. Discussion

The present study evaluates the efficacy of bovine lactoferrin (bLf) combined with sodium bicarbonate plus sodium citrate (Valpalf^®^) at a molar ratio of 10^−3^, in various biological activities. These include iron chelation, resistance to proteolysis, cellular internalization, its anti-inflammatory and antioxidant effects, and its potential as a therapeutic tool in AI in HT pregnant and non-pregnant women.

Iron chelation ability is a fundamental characteristic of bLf, crucial for its diverse biological functions, including anti-inflammatory, antioxidant, antibacterial, and antiviral ones [35,36]. Valpalf^®^ exhibits superior iron-binding capacity compared to nat-bLf when dissolved in sodium chloride, sodium citrate, or sodium bicarbonate. The results suggest that bLf in Valpalf^®^ enhances the iron saturation degree to 100%, highlighting its potential as a more effective iron chelator.

In accordance with the improved ability to bind iron, our study reveals that Valpalf^®^ displays enhanced resistance to proteolysis compared to nat-bLf, resembling the pattern of the resistance to degradation observed in holo-bLf, the iron-saturated glycoprotein. This finding indicates that Valpalf^®^, inducing a higher ability to chelate iron, confers stability to bLf, potentially prolonging its bioavailability and efficacy.

Moreover, the analysis of the cellular internalization and subcellular localization of bLf formulations in intestinal Caco-2 cells indicate that Valpalf^®^ exhibits efficient cellular entry and accumulation into the nucleus over time, suggesting its potential as an anti-inflammatory agent showing enhanced cellular uptake and activity compared to nat-bLf and holo-bLf.

In agreement with such potentiated iron-biding ability, resistance to proteolysis, and ability to accumulate into the cell nucleus, Valpalf^®^ shows higher anti-inflammatory and antioxidant properties. In particular, the results demonstrate that Valpalf^®^ effectively reduces IL-6 levels in LPS-stimulated macrophages and modulates the expression of antioxidant enzymes in PMA-stimulated monocytes, suggesting its potential to mitigate the oxidative stress-associated inflammation also caused by intracellular iron overload.

Finally, a retrospective study about the efficacy of Valpalf^®^ in the treatment of AI in HT pregnant and non-pregnant women demonstrates that Valpalf^®^ significantly improves hematological parameters, including RBCs, Hb, and TSI, excluding Ftn levels, while reducing IL-6 levels to a lower extent compared to bLf alone. Even if a larger sample size of anemic HT pregnant and non-pregnant women is needed for conclusive results, this is the first study that demonstrates the higher efficacy of bLf plus sodium bicarbonate and sodium citrate, compared with bLf alone, in treating AI.

Further research is warranted to elucidate the underlying mechanisms of the higher efficacy of Valpalf^®^ compared with native-bLf alone.

In this respect, some studies have highlighted the role of sodium citrate in improving ferric ions solubilization and bioavailability. In particular, in the presence of ferric ions, sodium citrate forms ferric citrate, as determined experimentally by Spiro and colleagues [37,38]. The ability of citrate to chelate ferric ions leads to the formation of a series of stable species in aqueous solution, which, in turn, plays an important role in iron solubilization, mobilization, and utilization of iron in biological systems [39]. Since ferric citrate is soluble over a broad range of pH, such as those found in the stomach, the intestine, and the duodenum (the main site of oral iron absorption), it will enhance the bioavailability of iron [39]. Concerning the presence of sodium bicarbonate, Schade et al. [40] first demonstrated that HCO_3_^−1^ or CO_3_^−2^ were implicated in the formation of iron complexes of human serum transferrin. Successively, spectroscopic studies and the 3D structure suggested that the HCO_3_^−1^ or CO_3_^−2^ anions bind to the arginine residue in both lobes of human Lf (Arg-121 in the N-lobe and Arg-465 in the C-lobe), thus neutralizing the positive charge of this amino acid [41,42]. The involvement of the anions in the iron coordination binding seems to be ideal for the reversible binding of iron [41], since the protonation of the HCO_3_^−1^ or CO_3_^−2^ anions is a probable primary step in the breakup of the iron site at low pH [43].

According to Spiro et al. [37,38], we hypothesize that sodium citrate solubilizes ferric iron forming ferric dicitrate which, in turn, dissociates in ferric monocitrate and sodium citrate. The metal-chelating species which reacts with the protein is likely the monocitrate–iron complex. Moreover, Phelps and Antonini [44] hypothesized that the rate of iron binding by ovotransferrin, another transferrin present in white eggs, could be dependent on the nature of the metal chelators employed, such as nitrilotriacetate and citrate complexes. As demonstrated by Hanudel and colleagues [45], ferric citrate has been approved as an iron supplement in curing iron deficiency anemia in patients with non-dialysis chronic kidney disease. The efficacy of ferric citrate is dependent on the presence of ferroportin, even if the specific mechanisms involved are still unknown [45].

Here, we first demonstrated that ferric citrate and sodium bicarbonate favor the formation of holo-bLf. On the basis of the data obtained, and the considerations defined above, we suggest the following scheme for the reaction of bLf with ferric citrate in the presence of bicarbonate:

Sodium citrate + ferric ions = ferric dicitrate

Ferric dicitrate = ferric citrate + mono citrate

bLf + ferric citrate = bLf-ferric citrate

bLf-ferric citrate + bicarbonate = bLf-ferric bicarbonate + citrate

Overall, this study provides some evidence supporting the higher therapeutic potential of Valpalf^®^ in the treatment of HT patients compared to bLf alone.

## 4. Materials and Methods

### 4.1. In Vitro Studies

#### 4.1.1. Reagents

Dulbecco’s modified Eagle’s medium (DMEM), RPMI 1640, Fetal Bovine Serum (FBS), and 0.25% Trypsin–EDTA solution were obtained from Sigma–Aldrich (Milan, Italy). The Bradford reagent was purchased from Bio-Rad Italia (Milan, Italy).

#### 4.1.2. Bovine Lactoferrin

The highly purified bLf was generously provided by Vivatis Pharma Italia s.r.l. (Saputo Dairy, Southbank, VC, Australia). The protein purity, assessed via SDS-PAGE and silver nitrate staining, was found to be around 99%. The Limulus Amebocyte assay (Pyrochrome kit, PBI International, Milan, Italy) revealed a low level of LPS contamination at 0.5 ± 0.06 ng/mg. The concentration of bLf solutions was determined using UV spectroscopy with an extinction coefficient of 15.1 (280 nm, 1% solution). The percentage of iron saturation was detected by spectrophotometric analysis at 468 nm, with an extinction coefficient of 0.540 for a 1% solution of a 100% saturated Lf. The sample was soluble, and the resulting solution was clear. At 468 nm, the OD of the original bLf sample, native-bLf (nat-bLf), in the absence of added ferric ions, was 0.050, corresponding to an iron saturation of 9.3%.

Holo-bLf was prepared by incubating a 20 mg/mL solution of the above-mentioned lot of bLf (2.5 × 10^−4^ M) in 0.8 × 10^−1^ M ferric citrate plus 1.7 × 10^−1^ M sodium bicarbonate for 30 min under stirring. The resulting holo-bLf underwent a 48 h dialysis against phosphate buffer solution (PBS) to remove unbound iron, sodium bicarbonate, and sodium citrate. The percentage of iron saturation of the obtained holo-bLf was detected by spectrophotometric analysis at 468 nm. The OD of the holo-bLf (1% solution) at 468 nm corresponded to 0.530, indicating an iron saturation of about 98%. The Holo-bLf was frozen and stored at −20 °C until utilized in experimental procedures.

In vitro experiments used nat-bLf and holo-bLf to compare the activities of bLf versus the same lot of bLf present in Valpalf^®^. Valpalf^®^ is a mixture described in Patent N. 102020000022420, priority 23 September 2020, containing bLf, sodium citrate, and sodium bicarbonate in the molar ratio (bLf moles/sodium citrate moles + sodium bicarbonate moles) ranging from 10^−3^ to 10^−4^. In the present study, the molar ratio of Valpalf^®^ (bLf/sodium citrate and sodium bicarbonate) corresponded to 1 × 10^−3^.

Before each in vitro assay, bLf solutions underwent sterilization using a 0.2 μm Millex HV filter with low protein retention (Millipore Corp., Bedford, MA, USA).

#### 4.1.3. Iron Titration Assay

The percentage of bLf iron saturation was detected by spectrophotometric analysis at 468 nm of 1% (10 mg/mL) solutions of nat-bLf (1.25 × 10^−4^ M), dissolved in different solutions such as 1.5 × 10^−1^ M sodium chloride, 4.0 × 10^−2^ M sodium citrate or 8.5 × 10^−2^ M sodium bicarbonate compared with 10 mg/mL bLf (1.25 × 10^−4^ M) plus 4.0 × 10^−2^ M sodium citrate and 8.5 × 10^−2^ M sodium bicarbonate solubilized in water. The molar ratio of Valpalf^®^ (bLf/sodium citrate and sodium bicarbonate) corresponded to 1 × 10^−3^. After dissolution, samples were soluble, and the resulting solutions were clear.

The concentration of the active protein, able to bind iron, was assessed by the repeated additions of 2.5 µL of 0.01 M of ferric ions, until equilibrium, followed by the OD lectures at 468 nm. After equilibrium was reached, a further addition of 25 µL of 0.01 M of ferric ions, followed by OD lectures at 468 nm, was performed.

#### 4.1.4. Tryptic Digestion

The samples of nat-bLf, holo-bLf, and Valpalf^®^ were solubilized in PBS at a concentration of 1 mg/mL (12.5 µM) and incubated with trypsin at a molar ratio of 12.5:1 (bLf:trypsin), within a time interval from 0 to 6 h at 37 °C. At different times of incubation (0, 10′, 30′, 1 h, 3 h, and 6 h), 20 µL from each solution was heat-treated at 90 °C for 3 min after the addition of SDS sample buffer containing dithiothreitol (DTT), and then stored at −20 °C. The tryptic digests were analyzed by SDS-PAGE and Coomassie blue staining.

#### 4.1.5. Cell Culture

The Caco-2 cells, derived from human colon carcinoma, were acquired from the American Type Culture Collection (ATCC), whereas the THP-1 cells were obtained from the European Collection of Cell Cultures (ECACC). Caco-2 cells were cultured in DMEM, while THP-1 cells were maintained in RPMI 1640 medium. For both cell lines, the medium was added with 10% FBS and 2 mM glutamine. The cells were cultured in a humidified incubator with 5% CO_2_ at 37 °C, ensuring a constant humid environment for their growth and viability. THP-1 cells, which showed spontaneous growth in suspension, were subcultured twice weekly. Subculture consists of pelleting and reseeding at a density of about 5 × 10^5^ cells/mL.

#### 4.1.6. Total and Nuclear Extracts

For internalization of Nat-, Holo-bLf, and Valpalf^®^, Caco-2 cells were seeded at a density of 7 × 10^5^ cells/well in 6-well tissue culture plates in complete DMEM medium for 48 h at 37 °C in a humidified incubator with 5% CO_2_. Caco-2 cells were washed with PBS and treated with 100 μg/mL of nat-bLf, holo-bLf, or Valpalf^®^ and harvested after 6 h and 24 h of treatment.

Total extracts were prepared by adding Lysis buffer (20 mM MOPS, NaCl 150 mM, Triton 1%, Leupeptin 2 µg/mL, Pepstatin 2 µg/mL, and PMSF 1 mM) to the cell pellets. The samples were then incubated at 4 °C for 30 min, with shaking every 10 min. After centrifugation at 13,000 rpm for 20 min at 4 °C, the supernatant was collected and stored at −80 °C in aliquots.

To prepare nuclear extracts, buffer A (containing 10 mM Hepes pH 7.9, 10 mM KCl, 1.5 mM MgCl_2_, 0.5 mM DTT, 0.1% NP40, and a 1:100 dilution of protease inhibitor cocktail) was added to the cell pellets to segregate nuclei from the cytosol. Following a 10 min incubation on ice, the samples underwent centrifugation at 11,000 rpm for 15 min at 4 °C. Following that, the pellets containing nuclear fractions were reconstituted in buffer C (consisting of 20 mM Hepes pH 7.9, 420 mM NaCl, 1.5 mM MgCl_2_, 25% glycerol, 1 mM EDTA, 1 mM EGTA, 0.5 mM DTT, 0.05% NP40, and a 1:50 dilution of protease inhibitor cocktail) and left to incubate on ice for 30 min. A final centrifugation at 13,000 rpm was then conducted, and the resulting supernatants were gathered and stored at −80 °C.

The total protein content was determined according to the Bradford method [46].

#### 4.1.7. Anti-Inflammatory Activity

THP-1 cells were seeded at a density of 2 × 10^6^ cells/well in 6-well tissue culture plates with RPMI medium (supplemented with 2 mM glutamine, 100 μM penicillin–streptomycin, and 10% FBS) and differentiated in macrophages by incubation with 0.16 μM PMA at 37 °C in 5% CO_2_ for 48 h. Differentiated THP-1 cells were washed with PBS and treated with 1 µg/mL LPS from *E. coli* (InvivoGen, USA). The stimulation was carried out for 48 h in an atmosphere of 95% air and 5% CO_2_ at 37 °C. A total of 100 μg/mL of nat-bLf, holo-bLf, or Valpalf^®^ were added to adherent THP-1 cells, without removing the culture medium, after 3 and 24 h of stimulation. After 48 h of incubation, the supernatants were harvested, aliquoted, and stored at −80 °C. IL-6 quantitation was carried out using the commercial kit Human ELISA Max Deluxe Sets (BioLegend, San Diego, CA, USA).

#### 4.1.8. Antioxidant Proteins

THP-1 cells were seeded at a density of 2 × 10^6^ cells/well in 6-well tissue culture plates with RPMI medium (supplemented with 2 mM glutamine, 100 μM penicillin–streptomycin, and 10% FBS), treated for 3 h with 100 µg/mL of nat-bLf, holo-bLf, or Valpalf^®^, and then challenged with 0.16 μM PMA. The cells were incubated at 37 °C in a humidified incubator with 5% CO_2_ for 24 h and then harvested and pelleted.

#### 4.1.9. Western Blot

Total protein quantitation was carried out by using the Bradford assay. Samples containing 20 µg of total protein in SDS sample buffer with DTT were heat-treated and loaded onto SDS-PAGE. Proteins were transferred onto nitrocellulose membranes (GE Healthcare, Life Sciences, Little Chalfont, Buckinghamshire, UK). The membranes were then incubated with primary antibodies diluted in PBS/5% non-fat dry milk/0.05% Tween-20 (Blotting-Grade Blocker, PanReac AppliChem, ITW Reagents, Monza, Italy) for 1 h at room temperature. The primary antibodies used included polyclonal anti-SOD-1 (sc-17767) (1:250), anti-SOD-2 (sc-137254) (1:500), monoclonal anti-bLf (sc-53498, Santa Cruz CA, USA) (1:1000), polyclonal anti-lamin A (Ab26300 Abcam; Milan, Italy), anti-tubulin (sc-23948) (1:1000), and GAPDH (sc-47724) (1:1000). After primary antibody incubation, the membranes were incubated with the appropriate HRP-conjugated secondary antibody (Bio-Rad, Milan, Italy) (1:1000) in PBS/5% non-fat dry milk/0.05% Tween-20 for 1 h at room temperature. Protein detection was performed using Clarity Western ECL substrate (170-5061, Bio-Rad, Milan, Italy). Discrepancies in protein levels among samples were normalized using tubulin as a reference for cytosolic extracts and lamin A for nuclear extracts.

### 4.2. Retrospective Study

#### 4.2.1. Study Design

To compare the safety and efficacy of the innovative Valpalf^®^ treatment (one capsule containing 200 mg of bLf + sodium citrate and bicarbonate, at a molar ratio of 10^−3^, two times a day before meals) versus the sole bLf treatment (one capsule containing 200 mg of bLf two times a day before meals) in curing AI in HT women, whether pregnant or not, a retrospective study was conducted. All HT women with AI were assessed by a gynecologist at Centro Diagnosi Medica, Via Regina Elena 270, Rome, Italy. Hematological parameters before and after Valpalf^®^ or bLf alone treatments were retrospectively collected. BLf has been marketed in Italy as a nutraceutical product for many years. Ethical approval was not deemed necessary, following the National Code on Clinical Trials declaration [47], as the data originated from real-life retrospective observations. All pregnant and non-pregnant women provided written informed consent for the use of the results obtained during the study. Additionally, the safety and tolerability of bLf or Valpalf^®^ were evaluated.

#### 4.2.2. Patients

Pregnant women from 18 to 40 years, between the 6th and 8th week of gestation, with a history of adverse outcomes including previous recurrent miscarriages, preterm birth, and intrauterine growth restriction were screened, before the treatment, for HT markers.

Non-pregnant women from 18 to 45 years were screened, before the treatment, for HT markers when the parents’ medical history revealed vascular diseases, such as heart attacks, strokes, intracranial hemorrhages, and the personal medical history was positive for heart attack, stroke, hypertension, venous vascular diseases, thrombophlebitis, previous pregnancies resulting in miscarriages, preeclampsia, eclampsia, intrauterine deaths, intrauterine growth restriction, and placental pathologies.

Exclusion criteria: pregnant and non-pregnant women with anti-phospholipid syndrome, other concomitant diseases, infections, previous iron supplementation therapy, recent blood transfusion(s), smokers, obese, and those allergic to milk proteins were excluded from this pilot study. Pregnant women with uncomplicated pregnancies were also excluded.

Inclusion criteria: pregnant and non-pregnant women suffering from HT and AI were considered in this retrospective study. HT pregnant women received low molecular-weight heparin (0.3 U/day of Seleparina, Italfarmaco SpA, Milano-Italy) and low-dose aspirin (100 mg every two days of Cardioaspirin^®^100, Bayer SpA, Milano-Italy) to prevent and reduce the risk of venous thromboembolism and miscarriage associated to hypercoagulability [48]. HT non-pregnant women received low-dose aspirin (100 mg every two days of Cardioaspirin^®^100, Bayer SpA, Milano-Italy) to prevent and reduce the risk of venous thromboembolism.

#### 4.2.3. Laboratory Analyses

HT markers were detected at Eurofins Genoma, Via Castel Giubileo, 11 00138 Rome, Italy, using the AnyplexTM real-time PCR kit (Seegene, Taewon Bldg. 91 Ogeum-ro, Songpa-gu, Seoul, Republic of Korea), following the protocol outlined in the brochure provided by AnyplexTM. In particular, both pregnant and non-pregnant women were screened for the following HT markers: protein C, activated protein C resistance, protein S, elevated coagulation factors, antithrombin deficiencies, hyper-homocysteinemia, F5 R506Q (factor V Leiden), and F2 G20210A (prothrombin G20210A). If at least one of these HT markers was detected, the women were considered to have HT and underwent further screening for hematological parameters before beginning treatment with bLf. Specifically, HT-positive pregnant and non-pregnant women received bLf or Valpalf^®^ treatment if one or more of the following hematological parameters met the specified values: RBCs ≤ 4000 × 10^3^/mL, Hb ≤ 11 g/dL, TSI ≤ 30 mg/dL, and sFtn ≤ 12 ng/mL. These hematological parameters were evaluated according to Meier et al. [49] both before starting treatment and after 30 days of treatment.

Blood IL-6 levels were quantified by ELISA Quantitative kits (R&D Systems, Wiesbaden, Germany) at the Department of Public Health and Infectious Diseases, Sapienza University of Rome, Italy, on the samples collected before and after 30 days of treatment.

#### 4.2.4. Study Population

From January 2022 to February 2024, a total of 134 HT women with AI were evaluated. Among these, 67 were HT pregnant women and 67 were HT non-pregnant women. All women were followed at the same center, the Centro di Diagnosi Medica, and the data were collected by the same gynecologist together with the other authors of the paper.

#### 4.2.5. Patients’ Treatments

A total of 35 HT pregnant women and 37 HT non-pregnant women took one capsule containing 200 mg of bLf twice a day (Mosiac^®^, Pharmaguida, Rome, Italy). Moreover, 32 HT pregnant women and 30 HT non-pregnant women received one capsule containing 200 mg of the same bLf plus sodium bicarbonate and sodium citrate at a molecular ratio of 10^−3^ (Valpalf^®^) (Mosiac Plus^®^, Pharmaguida, Rome, Italy), two times a day before meals. The purity of bLf, checked by SDS-PAGE and silver nitrate staining, was 99%. The iron saturation of bLf contained in the capsules was about 9.3%. At the time of data collection, all the patients had been taking bLf for 1 month. The mean values of hematological parameters collected, before and after the treatment, from these HT pregnant or non-pregnant women treated with 200 mg of bLf, were compared with the mean values found in HT pregnant and non-pregnant women treated with Valpalf^®^ two times a day before meals.

### 4.3. Statistical Analysis

Statistical analysis for experiments assessing Lf internalization, Lf digestion, as well as the antioxidant and anti-inflammatory activities of Lf was performed by using one-way ANOVA and Tukey’s post hoc test. The results, presented as the mean ± standard deviation (SD) from three independent experiments, considered a *p*-value ≤ 0.05 as statistically significant.

For the in vivo study, variables were summarized as mean ± SD. Hematological parameters are represented as values for each patient pre- and post-treatment, mean ± SD, and as percentage ± SD. The statistical analysis was carried out using ordinary one-way ANOVA with multiple comparisons. A *p*-value ≤ 0.05 was considered statistically significant.

Prism v7 software (GraphPad software, San Diego, CA, USA) was employed for statistical analysis.

## 5. Conclusions

In conclusion, the present study demonstrated that the addition of sodium bicarbonate and sodium citrate to a commercial nat-bLf, at a molar ratio of 10^−3^, potentiates the biological activity of the glycoprotein across the multifaceted spectrum of its multifunctionality, which could represent a novel nutraceutical and therapeutic agent with broad applications in health and disease management.

## Figures and Tables

**Figure 1 ijms-25-08559-f001:**
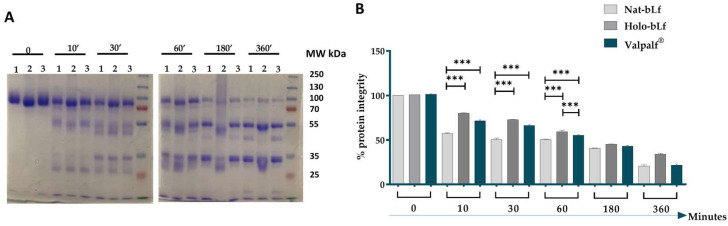
SDS-PAGE with Coomassie blue staining (**A**) and densitometry analysis (**B**) of nat-bLf (1), holo-bLf (2), and Valpalf^®^ (3) after different times of incubation with trypsin. For each lane, 6 µg of bLf were loaded. Densitometry analysis: the data are normalized on the 80 kDa band of nat-bLf. Statistical significance is indicated as follows: ***: *p* < 0.001 (one-way ANOVA with post hoc Tukey test).

**Figure 2 ijms-25-08559-f002:**
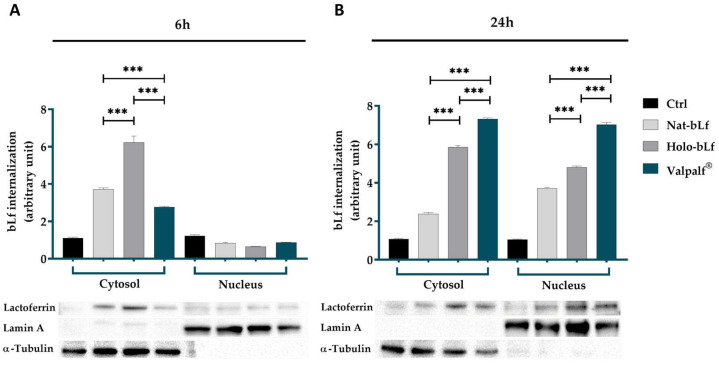
Analysis of bLf internalization and subcellular localization in Caco-2 cells. Western blot and densitometric analysis of bLfs in cytosolic and nuclear fractions after 6 h (**A**) and 24 h (**B**) of treatment with 100 μg/mL nat-bLf, holo-bLf or Valpalf^®^. The data, normalized to the internal housekeeping genes (α-Tubulin for cytosolic fraction and Lamin A for nuclear fraction), are presented as means ± SEM. Statistical significance is indicated as follows: ***: *p* < 0.001 (one-way ANOVA with post hoc Tukey test).

**Figure 3 ijms-25-08559-f003:**
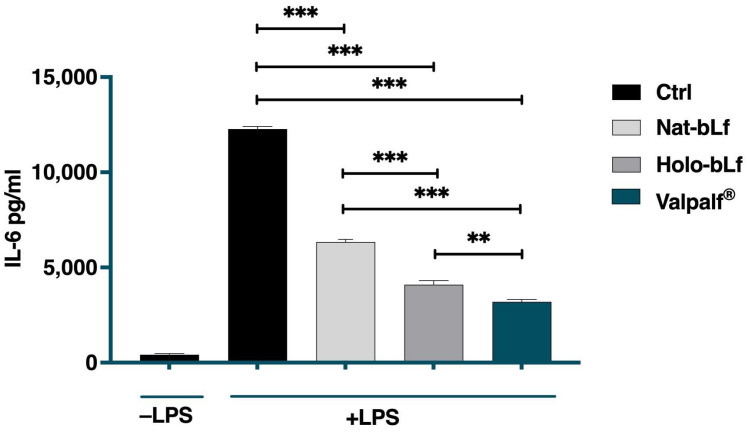
ELISA quantitation of IL-6 levels in THP-1 cells untreated or challenged with 1 µg/mL LPS in the absence or presence of 100 μg/mL nat-bLf, holo-bLf, or Valpalf^®^. Error bars: standard error of the mean. Statistical significance is indicated as follows: **: *p* < 0.01; ***: *p* < 0.001 (one-way ANOVA with post hoc Tukey test).

**Figure 4 ijms-25-08559-f004:**
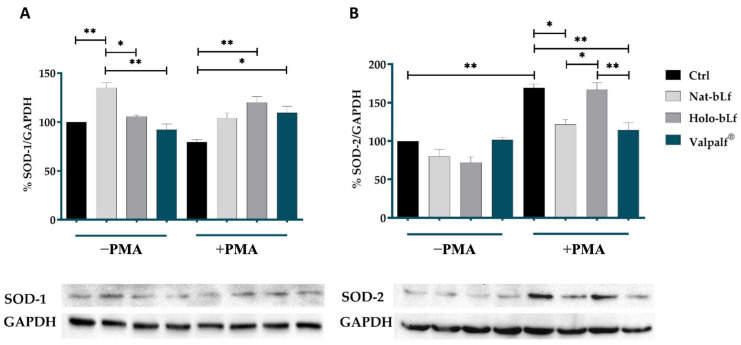
Western blot and densitometry analysis of SOD-1 (**A**) and SOD-2 (**B**) levels in THP-1 cells undifferentiated or differentiated with 0.16 µM PMA and treated with 100 µg/mL of nat-bLf, holo-bLf, or Valpalf^®^. Data were calculated relative to the housekeeping gene GAPDH. Error bars: standard error of the mean. Statistical significance is indicated as follows: *: *p* < 0.05; **: *p* < 0.01 (one-way ANOVA with post hoc Tukey test).

**Figure 5 ijms-25-08559-f005:**
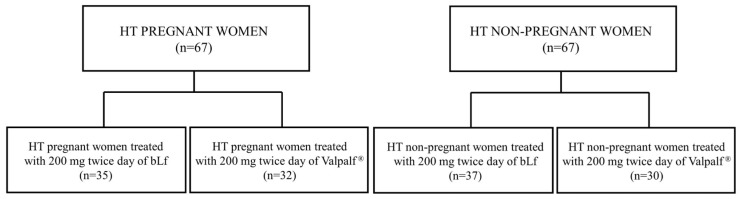
Flow diagrams of enrolled pregnant and non-pregnant women suffering from hereditary thrombophilia (HT) and treated with bLf alone or Valpalf^®^.

**Figure 6 ijms-25-08559-f006:**
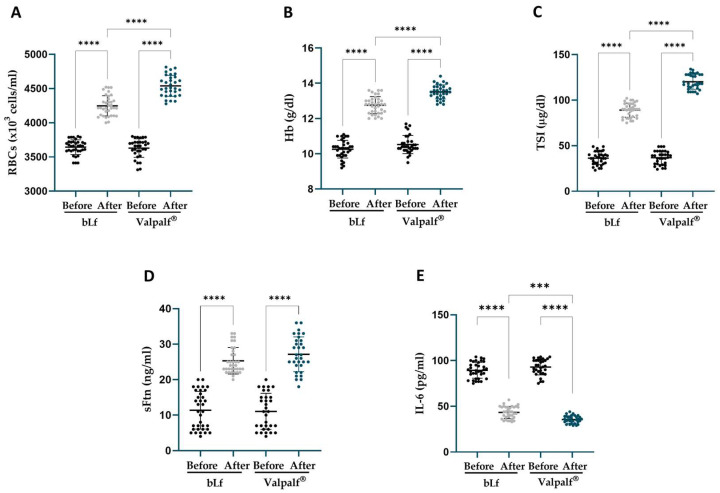
Levels of red blood cells (RBCs) (**A**), hemoglobin (Hb) (**B**), total serum iron (TSI) (**C**), serum ferritin (sFtn) (**D**) and IL-6 (**E**) before (black dots) and after 30 days of treatment with 200 mg two times a day of bLf (light grey dots) in 35 HT pregnant women or 200 mg two times a day of Valpalf^®^ (blue dots) in 32 HT pregnant women. Dots represented the values for each patient pre- and post-treatments. Mean values ± standard deviation are also reported. Statistical significance is indicated as follows: ***: *p* < 0.001; ****: *p* < 0.0001 (one-way ANOVA with multiple comparisons).

**Figure 7 ijms-25-08559-f007:**
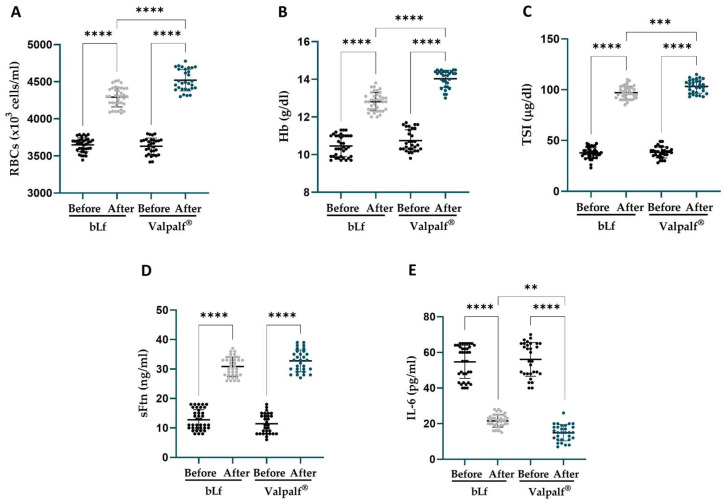
Levels of red blood cells (RBCs) (**A**), hemoglobin (Hb) (**B**), total serum iron (TSI) (**C**), serum ferritin (sFtn) (**D**) and IL-6 (**E**) before (black dots) and after 30 days of treatment with 200 mg two times a day of bLf (light grey dots) in 37 HT non-pregnant women or 200 mg two times a day of Valpalf^®^ (blue dots) in 30 HT non-pregnant women. Dots represented the values for each patient pre- and post-treatments. Mean values ± standard deviation are also reported. Statistical significance is indicated as follows: **: *p* < 0.01; ***: *p* < 0.001; ****: *p* < 0.0001 (one-way ANOVA with multiple comparisons).

**Table 1 ijms-25-08559-t001:** Iron titration with 0.01 M ferric chloride of 1.25 × 10^−4^ M bLf in 1.5 × 10^−1^ M sodium chloride or in 4.0 × 10^−2^ M sodium citrate or in 8.5 × 10^−2^ M sodium bicarbonate compared to Valpalf^®^ in distilled water (dH_2_O).

Reagent	OD AT 468 nm (% Iron Saturation)			
Ferric Chloride	bLf in Sodium Chloride	bLf in Sodium Citrate	bLf in Sodium Bicarbonate	Valpalf^®^ in dH_2_O
**-**	0.050 (9.3)	0.052 (9.6)	0.051 (9.4)	0.050 (9.3)
**2.5 µL**	0.068 (12.6)	0.067 (12.4)	0.069 (12.8)	0.105 (19.4)
**2.5 µL**	0.083 (15.4)	0.098 (18.1)	0.099 (18.3)	0.168 (31.1)
**2.5 µL**	0.096 (17.8)	0.107 (19.8)	0.103 (19.1)	0.256 (47.4)
**2.5 µL**	0.103 (19.1)	0.115 (21.3)	0.114 (21.1)	0.330 (61.1)
**2.5 µL**	0.110 (20.4)	0.122 (22.6)	0.129 (23.9)	0.411 (76.1)
**2.5 µL**	0.123 (22.8)	0.131 (24.3)	0.131 (24.3)	0.470 (87.0)
**2.5 µL**	0.127 (23.5)	0.139 (25.7)	0.135 (25.0)	0.498 (92.2)
**2.5 µL**	0.142 (26.3)	0.146 (27.0)	0.142 (26.3)	0.540 (100)
**25 µL**	0.151 (**28.0**)	0.157 (**29.1**)	0.158 (**29.3**)	0.540 (**100**)

**Table 2 ijms-25-08559-t002:** Characteristics of 35 and 32 hereditary thrombophilic (HT) pregnant women suffering from anemia of inflammation before bLf and Valpalf^®^ treatment, respectively.

Parameters	HT Pregnant Women (*n* = 35) before bLf Treatment	HT Pregnant Women (*n* = 32) before Valpalf^®^ Treatment	*p*-Value
**Age**	28.6 ± 6.3	28.4 ± 7.1	0.9989
**RBCs (×10^3^ cells)**	3643 ± 107.9	3630 ± 136	0.9762
**Hb (g/dL)**	10.3 ± 0.5	10.5 ± 0.5	0.1188
**TSI (μg/dL)**	36.3 ± 7.0	36.8 ± 7.4	0.9933
**sFtn (ng/mL)**	11.4 ± 5.3	11.1 ± 5.1	0.9935
**IL-6 (pg/mL)**	89.4 ± 8.6	92.9 ± 8.5	0.1987

Data are reported as mean ± standard deviation. RBCs = red blood cells; Hb = hemoglobin; TSI = total serum iron; sFtn = serum ferritin; IL-6 = interleukin-6.

**Table 3 ijms-25-08559-t003:** Characteristics of 37 and 30 hereditary thrombophilic (HT) non-pregnant women suffering from anemia of inflammation before bLf and Valpalf^®^ treatment, respectively.

Parameters	HT Non-Pregnant Women (*n* = 37) before bLf Treatment	HT Non-Pregnant Women (*n* = 30) before Valpalf^®^ Treatment	*p*-Value
**Age**	36 ± 9	37.1 ± 8.1	0.9703
**RBCs (×10^3^ cells)**	3649 ± 93.5	3631 ± 111.1	0.9215
**Hb (g/dL)**	10.5 ± 0.6	10.7 ± 0.6	0.1212
**TSI (μg/dL)**	37.8 ± 5.6	38.3 ± 5.4	0.9855
**sFtn (ng/mL)**	12.8 ± 3.4	11.4 ± 3.4	0.3851
**IL-6 (pg/mL)**	54.7 ± 9.3	56.1 ± 9.5	0.8662

Data are reported as mean ± standard deviation. RBCs = red blood cells; Hb = hemoglobin; TSI = total serum iron; sFtn = serum ferritin; IL-6 = interleukin-6.

## Data Availability

The data presented in this study are available in the current article.

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
