# Peer review of "Valpalf®: A New Nutraceutical Formulation Containing Bovine Lactoferrin That Exhibits Potentiated Biological Activity"

_ijms, 2024, doi:10.3390/ijms25168559_

Round 1
Reviewer 1 Report (Previous Reviewer 1)
Comments and Suggestions for Authors
My comment on ijms-3133913 - review after correction.
The manuscript has been sufficiently improved and now warrants publication in ijms, but minor corrections are required.
Regarding point 7, we did not find any note on Fig. 4.
I don't know why my comment was lost, but I meant the difference in the notation SOD-1 and SOD-2 in the text and in Fig. 4 (once it is in brackets and once without brackets),
e.g. (SOD)-1 and SOD-2 was assessed through Western blot analysis (Figure 4), see line 220,
or e.g. The expression of SOD-1 was significantly higher in undifferentiated cells treated, see line 221and see caption of Fig. 4. see line 241.
See line 395 - - the citation position is 44, not 4 (Moreover, Phelps and Antonini [441] …
Author Response
We thank the Reviewer for his/her comments. The manuscript has been amended accordingly, and a point-to-point reply follows. The new revisions are displayed in track change format.
The manuscript has been sufficiently improved and now warrants publication in IJMS, but minor corrections are required.
Regarding point 7, we did not find any note on Fig. 4.
I don't know why my comment was lost, but I meant the difference in the notation SOD-1 and SOD-2 in the text and in Fig. 4 (once it is in brackets and once without brackets),
e.g. (SOD)-1 and SOD-2 was assessed through Western blot analysis (Figure 4), see line 220,
or e.g. The expression of SOD-1 was significantly higher in undifferentiated cells treated, see line 221and see caption of Fig. 4. see line 241.
The text has been corrected to avoid differences in the notation of SOD-1 and SOD-2.
See line 395 - - the citation position is 44, not 4 (Moreover, Phelps and Antonini [441] …
Done.
Reviewer 2 Report (New Reviewer)
Comments and Suggestions for Authors
This is a complete and very interesting research about the functional properties of a product containing lactoferrin (Valpalf®), compared to native and holo-lactoferrin. In general, the research is well done. I have several issues:
1. Fig. 1B. It is not apparent that Valpalf®showed more resistance to trypsin than holo-Lf.
2. Fig. 1. Please add the molecular weights of markers.
3. The expression of SOD-1 is major when holo-Lf is applied.
4. Why if holo-Lf had better results than native Lf you used native Lf (9.2% iron saturation) for the experiments of treatments to patients.
5. Data of Tables 2 and 3 are of before treatments, and after treatments are lacking.
6. Please explain why did you use 400 mg/day of Lf given to patients? Do you have these data from other assays?
Author Response
We thank the Reviewer for his/her comments. The manuscript has been amended accordingly, and a point-to-point reply follows. The new revisions are displayed in track change format.
- - 1B. It is not apparent that Valpalf® showed more resistance to trypsin than holo-Lf.
The text never states that Valpalf® is more resistant to trypsin than holo-bLf, but that it is rather more resistant to trypsin than nat-bLf (lines 157-168 of the pdf file). There was also a typo at the beginning of the paragraph (ability instead of mobility), we corrected it.
- 1. Please add the molecular weights of markers.
Done.
- The expression of SOD-1 is major when holo-Lf is applied.
Thanks for the comment. We do not see a higher effect of holo-bLf on SOD-1. Indeed, variations in the expression of SOD-1 between PMA-untreated and PMA-treated cells do not appear to be significant, regardless of the form of bLf applied, suggesting that, under these conditions, this enzyme is poorly regulated. In contrast, SOD-2 appears to act as a pro-oxidant stress sensor, as demonstrated by its critical up-expression following PMA treatment. The effects of nat-bLf, and even more so of Valpalf® treatment, in reversing this increase are highly significant, suggesting that both can enhance their antioxidant activity through direct chelation of free iron. Holo-bLf is not efficient in this task, highlighting functional limitations in its application.
- Why if holo-Lf had better results than native Lf you used native Lf (9.2% iron saturation) for the experiments of treatments to patients.
The retrospective studies were intended to compare the effect of Valpalf® (which contains nat-bLf) to that of nat-bLf alone. It should be kept in mind, in any case, that, under inflammatory conditions or in iron homeostasis disorders, patients have iron overload (Rosa et al 2017), so it is recommended to administer a native bLf preparation with low iron saturation that can chelate the excess of iron. Of note, this fundamental iron-chelating function exerted by native bLf is enhanced by the addition of sodium bicarbonate and citrate, as demonstrated by the major iron binding ability of the new formulation Valpalf®.
- Data of Tables 2 and 3 are of before treatments, and after treatments are lacking.
The data in Tables 2 and 3 simply report the baselines values of both hereditary thrombophilic (HT) non-pregnant and pregnant women suffering from anemia of inflammation before bLf and Valpalf® treatment. Data after treatments are reported in Figures 6 and 7.
- Please explain why did you use 400 mg/day of Lf given to patients? Do you have these data from other assays?
Different studies on anemic pregnant woman treated with 100 mg bLf twice a day have reported conflicting results, when compared to standard treatment based on iron supplementation, regarding the significant increase of hemoglobin, but with the advantage of the absence of iron-related side effects. In particular, some studies highlighted significant differences between bLf and iron-based treatments (Paesano et al., 2006; Paesano et al., 2010; Paesano et al., 2014; Rezk et al., 2016; Lepanto et al. 2018; Fawzy Mohamed et al., 2020; Bayoumy et al., 2021), whereas others evidenced no statistical difference among them (Nappi et al., 2009; Ali et al., 2015; Darwish et al., 2018).
Consequently, we increased the quantity of bLf from 100 to 200 mg twice a day. The increase in hematological values observed upon 200 mg of nat-bLf administered twice a day is higher compared to the data obtained by administering 100 mg nat-bLf twice a day. Such increase is even higher when administering 200 mg twice a day of the new formulation Valpalf®.
This manuscript is a resubmission of an earlier submission. The following is a list of the peer review reports and author responses from that submission.
Round 1
Reviewer 1 Report
Comments and Suggestions for Authors Thank you for your questions and comments. I am sending more detailed answers and comments. The main questions addressed in the study are: 1. Have the studies shown that the addition of Valpalf® to commercial nat-bLf enhances the biological activity of the glycoprotein in the entire spectrum of its multifunctionality? I think so. 1a. Whether it represents a new broad-spectrum nutraceutical and therapeutic agent with applications in health and disease management. I believe that this combination of commercial nat-bLf with the addition of Valpalf® (based on the described studies) may constitute a new therapeutic agent. 2. I consider it original and important to verify that iron citrate and sodium bicarbonate help improve the solubility and bioavailability of iron ions and bind to holo-bLf. 2a. The study provides comprehensive evidence confirming the multi-aspect biological effects and therapeutic potential of bLf plus Valpalf® in various physiological and pathological conditions, which fills the gap in the field covered by the article. 3. The study provides a wide spectrum of in vitro experiments in the field of iron chelation, resistance to proteolysis, cellular internalization, anti-inflammatory and antioxidant effects. 3a. Finally, a retrospective (in vivo) study on the effectiveness of bLf plus Valpalf® in the treatment of AI in pregnant and non-pregnant women with HT shows that bLf plus Valpalf® significantly improves hematological parameters, including RBC, Hb, TSI, excluding Ftn, with while reducing IL-6 levels to a lesser extent compared to bLf alone. This highlights the potential synergistic effect of bLf and Valpalf® in alleviating inflammation-related anemia. 4. In further studies (in vitro), you can also try to perform tests with other gastrointestinal cell lines than Caco-2 cells (maybe with normal lines as a control). 4a. Further studies are recommended to elucidate the mechanisms underlying the superior efficacy of bLf plus Valpalf® compared to native bLf alone. 5. The conclusions are consistent with the evidence and arguments presented. They are based on the experiments performed and the presented results as well as the cited literature. 5a. The iron titration test (Table 1) showed that the degree of iron saturation is approximately 28-29% for all Nat-bLf solutions, with the notable exception of bLf plus Valpalf®, which is the only one that achieves 100% iron saturation. thus demonstrating its higher iron-binding capacity compared to Nat-bLf dissolved in sodium citrate alone or sodium bicarbonate, at the same concentrations used in bLf plus Valpalf®, where citrate and sodium bicarbonate are added simultaneously. 5b. All further experiments were then performed comparing nat-bLf, holo-bLf and bLf plus Valpalf® in terms of different biological activities. Studies were carried out in the field of iron chelation, resistance to proteolysis, cellular internalization, anti-inflammatory and antioxidant effects (in vitro) and a retrospective study was performed: the effectiveness of bLf plus Valpalf® in the treatment of inflammatory anemia in pregnant and non-pregnant women with hereditary thrombophilia (Fig. 6-7), which respond to the research issues presented. 6. The quality of data in tables and figures is adequate. The references are appropriate, but need to be corrected in order - I wrote about this earlier. 7. I posted a note to Fig. 4 regarding (SOD)-1 earlier.Author Response
Thank you for your questions and comments. I am sending more detailed answers and comments. The main questions addressed in the study are: 1. Have the studies shown that the addition of Valpalf® to commercial nat-bLf enhances the biological activity of the glycoprotein in the entire spectrum of its multifunctionality? I think so. 1a. Whether it represents a new broad-spectrum nutraceutical and therapeutic agent with applications in health and disease management. I believe that this combination of commercial nat-bLf with the addition of Valpalf® (based on the described studies) may constitute a new therapeutic agent. 2. I consider it original and important to verify that iron citrate and sodium bicarbonate help improve the solubility and bioavailability of iron ions and bind to holo-bLf. 2a. The study provides comprehensive evidence confirming the multi-aspect biological effects and therapeutic potential of bLf plus Valpalf® in various physiological and pathological conditions, which fills the gap in the field covered by the article. 3. The study provides a wide spectrum of in vitro experiments in the field of iron chelation, resistance to proteolysis, cellular internalization, anti-inflammatory and antioxidant effects. 3a. Finally, a retrospective (in vivo) study on the effectiveness of bLf plus Valpalf® in the treatment of AI in pregnant and non-pregnant women with HT shows that bLf plus Valpalf® significantly improves hematological parameters, including RBC, Hb, TSI, excluding Ftn, with while reducing IL-6 levels to a lesser extent compared to bLf alone. This highlights the potential synergistic effect of bLf and Valpalf® in alleviating inflammation-related anemia. 4. In further studies (in vitro), you can also try to perform tests with other gastrointestinal cell lines than Caco-2 cells (maybe with normal lines as a control). 4a. Further studies are recommended to elucidate the mechanisms underlying the superior efficacy of bLf plus Valpalf® compared to native bLf alone. 5. The conclusions are consistent with the evidence and arguments presented. They are based on the experiments performed and the presented results as well as the cited literature. 5a. The iron titration test (Table 1) showed that the degree of iron saturation is approximately 28-29% for all Nat-bLf solutions, with the notable exception of bLf plus Valpalf®, which is the only one that achieves 100% iron saturation. thus demonstrating its higher iron-binding capacity compared to Nat-bLf dissolved in sodium citrate alone or sodium bicarbonate, at the same concentrations used in bLf plus Valpalf®, where citrate and sodium bicarbonate are added simultaneously. 5b. All further experiments were then performed comparing nat-bLf, holo-bLf and bLf plus Valpalf® in terms of different biological activities. Studies were carried out in the field of iron chelation, resistance to proteolysis, cellular internalization, anti-inflammatory and antioxidant effects (in vitro) and a retrospective study was performed: the effectiveness of bLf plus Valpalf® in the treatment of inflammatory anemia in pregnant and non-pregnant women with hereditary thrombophilia (Fig. 6-7), which respond to the research issues presented. 6. The quality of data in tables and figures is adequate. The references are appropriate, but need to be corrected in order - I wrote about this earlier. 7. I posted a note to Fig. 4 regarding (SOD)-1 earlier.
We thank the Reviewer for his/her comments.
We apologize for incorrectly defining the Valpalf® brand. In fact, Valpalf® is a brand that refers to a mixture that includes bLf, sodium bicarbonate, and sodium citrate. Therefore, it is sufficient to indicate the formulation just as “Valpalf” instead of “bLf plus Valpalf”. The manuscript, along with the title, has been amended accordingly.
Regarding points 4 and 4a, we certainly foresee to continue our studies on other gastrointestinal cell lines in order to elucidate the mechanism beyond the superior efficacy of Valpalf® vs bLf alone.
Regarding point 6, references have been correctly ordered.
Regarding point 7, we did not find any note on Fig. 4.
Reviewer 2 Report
Comments and Suggestions for Authors
Although the topic discussed is important for public health as well as for hereditary thrombophilic pregnant and non-pregnant women the presented manuscript is burdened with numerous errors.
The first limitation of the study, for in vivo section, is the lack of detailed characterisation of cohorts studied. The including and excluding criteria are missing as well as control group, namely patients without hereditary thrombophilic disorder. If this factor that dominates work has not been taken into account, the conducted research does not allow drawing substantive conclusions. Did the patients take, for example, anti-inflammatory drugs or other drugs, e.g. supplements during pregnancy? There is no additional information regarding such inclusion and exclusion parameters for the study.
Moreover, the details concerning analyzed cohorts are provided for only two, but must be provided for all four analyzed cohorts with dedicated statistical verifications, to confirm whether there is any influence of confounding factors.
Also, why are only mean values ± standard deviation shown (refers to Figures 6 and 7)? In this case, wouldn't it be appropriate to analyze the changes for individuals more carefully? Perhaps changes should be presented in the form of % for a more advanced statistical analysis.
Taken together, the lack of control group and detailed characteristics of the study groups makes it impossible to reliably evaluate the obtained results.
Additionally, there are doubts as to whether this is a dedicated journal for presenting research results regarding the use of a specific preparation? In my opinion, pharmaceutical journals would be more appropriate.
Introduction - Line 88-90
The sentence „Numerous in vitro [15,23], in vivo [24,25], and clinical [26,27] studies have highlighted the potent anti-inflammatory properties of bovine Lf (bLf) through various mechanisms…” requires enrichment by adding appropriate citations.
Table 1 (a) concentrations of salts should be provided using the same concentration unit, (b) the values of initial saturations should be provided.
Table 2. Characteristics of hereditary thrombophilic (HT) pregnant and non-pregnant women suffering from anemia of inflammation.
Requires statistical verification - is there no statistical difference between the groups? Moreover, it is necessary to show the values for the analyzed parameters for 4 groups according to Figure 5. Flow diagrams of enrolled pregnant (A) and non-pregnant (B) women. Were the analyzed groups homogeneous?
Discussion - Line 355 – „This highlights the potential synergistic effects of bLf plus Valpalf® in alleviating inflammation-associated anemia” - the statement is an overinterpretation
and the same applied to
Line 394 – „Overall, the study provides comprehensive evidence supporting the multifaceted biological activities and therapeutic potential of bLf plus Valpalf® across various physiological and pathological conditions.”
Materials - Line 427
„BLf plus Valpalf® is a mixture, described in granted Patented N. 102020000022420, priority 23 September 2020, containing bLf, sodium citrate and sodium bicarbonate in the molar ratio (bLf moles/sodium citrate moles + sodium bicarbonate moles) ranging from 10-3 to 10-4.”
Concentration should be given as a specific value. Lack of Lf, sodium citrate and sodium bicarbonate concentrations - it is impossible to assess its influence on the results obtained. Moreover, the iron saturation and purity of BLf plus Valpalf® is missing.
Minor
Abbreviations used should be standardized throughout the work, e.g. Nat-bLf/nat-bLf
Line 164 should be corrected to numer (Rosa et al. 2018)
Line 172 – repetition Figure 1. Figure 1
Numbering of subchapters - twice 2.3.
Figure 3 – custom description of the y axis
Comments on the Quality of English LanguageMinor editing of English language required
Author Response
Although the topic discussed is important for public health as well as for hereditary thrombophilic pregnant and non-pregnant women the presented manuscript is burdened with numerous errors.
We thank the Reviewers for the helpful comments. We apologize for incorrectly defining the Valpalf® brand. In fact, Valpalf® is a brand that refers to a mixture that includes bLf, sodium bicarbonate, and sodium citrate. Therefore, it is sufficient to indicate the formulation just as “Valpalf” instead of “bLf plus Valpalf”. The manuscript, along with the title, has been amended accordingly, and a point-to-point reply follows.
The first limitation of the study, for in vivo section, is the lack of detailed characterisation of cohorts studied. The including and excluding criteria are missing as well as control group, namely patients without hereditary thrombophilic disorder. If this factor that dominates work has not been taken into account, the conducted research does not allow drawing substantive conclusions. Did the patients take, for example, anti-inflammatory drugs or other drugs, e.g. supplements during pregnancy? There is no additional information regarding such inclusion and exclusion parameters for the study.
We agree with the Reviewer, therefore inclusion and exclusion criteria are now reported in the revised version of the manuscript (section 4.2.2 lines 567-585 of the pdf file). Regarding the treatment of non-HT women (control group), it should be remembered that the imbalance of serum parameters, such as red blood cell count, hemoglobin, total serum iron, serum ferritin and IL-6, is typically found in patients suffering from HT or other pathologies, but is normally not detected in healthy subjects. The effectiveness of bLf in improving serum parameters in non-healthy patients (including HT patients) has already been demonstrated in previous studies (Paesano et al. 2014; Lepanto et al. 2018). The present study was focused on comparing the efficacy of a new formulation containing bLf (Valpalf®) vs. bLf alone in relieving HT patients from inflammation-induced anemia (AI).
Moreover, the details concerning analyzed cohorts are provided for only two, but must be provided for all four analyzed cohorts with dedicated statistical verifications, to confirm whether there is any influence of confounding factors.
The details and the statistical analysis of all four cohorts have been included in Tables 2 and 3. As the statistical analysis shows, there were no significant differences between the sub-cohorts undergoing different treatments (bLf alone vs Valpalf®).
Also, why are only mean values ± standard deviation shown (refers to Figures 6 and 7)? In this case, wouldn't it be appropriate to analyze the changes for individuals more carefully? Perhaps changes should be presented in the form of % for a more advanced statistical analysis.
Figure 6 and 7 already contains the values of each patient included in the study. However, as suggested by the Reviewer, changes are now also reported as % in the text (lines 288-320 of the pdf file).
Taken together, the lack of control group and detailed characteristics of the study groups makes it impossible to reliably evaluate the obtained results.
The groups are now thoroughly detailed. See above for the lack of control healthy subjects.
Additionally, there are doubts as to whether this is a dedicated journal for presenting research results regarding the use of a specific preparation? In my opinion, pharmaceutical journals would be more appropriate.
Obviously this is a matter for the Editor, but we believe the article is fully within the scope of the Journal.
Introduction - Line 88-90
The sentence „Numerous in vitro [15,23], in vivo [24,25], and clinical [26,27] studies have highlighted the potent anti-inflammatory properties of bovine Lf (bLf) through various mechanisms…” requires enrichment by adding appropriate citations.
Other references have been added as requested.
Table 1 (a) concentrations of salts should be provided using the same concentration unit, (b) the values of initial saturations should be provided.
Table 1 now reports saturation values at all points.
Table 2. Characteristics of hereditary thrombophilic (HT) pregnant and non-pregnant women suffering from anemia of inflammation.
Requires statistical verification - is there no statistical difference between the groups? Moreover, it is necessary to show the values for the analyzed parameters for 4 groups according to Figure 5. Flow diagrams of enrolled pregnant (A) and non-pregnant (B) women. Were the analyzed groups homogeneous?
Details of the four groups have been added into Tables 2 and 3, and statistical analysis has evidenced no significant difference between the two sub-cohorts undergoing different treatments (bLf alone vs Valpalf®).
Discussion - Line 355 – „This highlights the potential synergistic effects of bLf plus Valpalf® in alleviating inflammation-associated anemia” - the statement is an overinterpretation
and the same applied to
Line 394 – „Overall, the study provides comprehensive evidence supporting the multifaceted biological activities and therapeutic potential of bLf plus Valpalf® across various physiological and pathological conditions.”
Both sentences have been amended accordingly.
Materials - Line 427
“BLf plus Valpalf® is a mixture, described in granted Patented N. 102020000022420, priority 23 September 2020, containing bLf, sodium citrate and sodium bicarbonate in the molar ratio (bLf moles/sodium citrate moles + sodium bicarbonate moles) ranging from 10-3 to 10-4.”
Concentration should be given as a specific value. Lack of Lf, sodium citrate and sodium bicarbonate concentrations - it is impossible to assess its influence on the results obtained. Moreover, the iron saturation and purity of BLf plus Valpalf® is missing.
The amount of bLf is clearly stated into the text, each capsule contains 200 mg of the protein, sodium bicarbonate and sodium citrate at a molar ratio of 10-3 (lines 548-553, 613-617 of the pdf file).
Iron saturation is reported in Table 1 and purity in M&M section, where it is detailed that Valpalf® was prepared by using the same lot of bLf, whose purity had been assessed by SDS-PAGE and silver nitrate staining and found to be around 99% (lines 424-433, 618-619 of the pdf file).
Minor
Abbreviations used should be standardized throughout the work, e.g. Nat-bLf/nat-bLf
Line 164 should be corrected to number (Rosa et al. 2018)
Line 172 – repetition Figure 1. Figure 1
Numbering of subchapters - twice 2.3.
Figure 3 – custom description of the y axis
The text has been amended accordingly.